# Using the Conditional Process Analysis Model to Characterize the Evolution of Carbon Structure in *Taxodium ascendens* Biochar with Varied Pyrolysis Temperature and Holding Time

**DOI:** 10.3390/plants13030460

**Published:** 2024-02-05

**Authors:** Shuai Zhang, Xiangdong Jia, Xia Wang, Jianyu Chen, Can Cheng, Xichuan Jia, Haibo Hu

**Affiliations:** 1College of Forestry, Nanjing Forestry University, Nanjing 210037, China; fuji@njfu.edu.cn (S.Z.); xiawang1019@126.com (X.W.); 2180100042@njfu.edu.cn (J.C.); cheng-can1991@njfu.edu.cn (C.C.); jiaxichuan@njfu.edu.cn (X.J.); 2Co-Innovation Center of Sustainable Forestry in Southern China, Nanjing 210037, China; 3College of Science, Nanjing Forestry University, Nanjing 210037, China; jxd_nld0401@sina.com

**Keywords:** aromatic structure, biochar stability, holding time, branch and leaf feedstocks, conditional process analysis

## Abstract

Temperature determines biochar structure during pyrolysis. However, differences in holding time and feedstock types may affect this relationship. The conditional process analysis model was used in this paper to investigate the potential to affect this mechanism. The branch and leaf parts of *Taxodium ascendens* were separately pyrolyzed at 350, 450, 650, and 750 °C, and kept for 0.5, 1, and 2 h at each target temperature. We measured the fixed carbon and ash contents and the elemental composition (C, H, O and N) of the raw materials and their char samples. After plotting a Van Krevelen (VK) diagram to determine the aromatization of chars, the changes in the functional groups were analyzed using Fourier transform infrared (FTIR), Raman, and X-ray photoelectron spectroscopy (XPS). The results revealed that pyrolysis at temperatures between 450 and 750 °C accounted for the aromatization of biochar because the atomic H/C ratio of branch-based chars (BC) decreased from 0.53–0.59 to 0.15–0.18, and the ratio of leaf-based chars (LC) decreased from 0.56–0.68 to 0.20–0.22; the atomic O/C ratio of BC decreased from 0.22–0.27 to 0.08–0.11, while that of LC decreased from 0.26–0.28 to 0.18–0.21. Moreover, the average contents of N (1.89%) and ash (13%) in LC were evidently greater than that in BC (N:0.62%; Ash: 4%). Therefore, BC was superior to LC in terms of the stability of biochar. In addition, the increasing I_D_/I_G_ and I_D_/I_(DR+GL)_ ratios in BC and LC indicated an increasing amount of the amorphous aromatic carbon structure with medium-sized (2~6 rings) fused benzene rings. According to the CPA analysis, an extension of the holding time significantly enhanced the increase in aromatic structures of LC with temperature. But this extension slightly reduced the growth in aromatic structures of BC. All indicate that holding time and feedstock types (branch or leaf feedstock) could significantly affect the variation in biochar aromatic structure with respect to temperature.

## 1. Introduction

Studies have demonstrated the positive and pivotal roles of biochar in forestalling C degradation and increasing the C sink for an entire ecosystem [1,2,3,4]. The polycondensation caused by high-temperature pyrolysis converts the organic carbon in biomasses to aromatic carbon; this accounts for the stability of biochar [5,6,7]. The aromaticity and stability of biochar are directly related to the content of C, H, O, and N [8]. Investigations have shown that the concentration of aromatic C in biochar increases with decreasing H/C and O/C ratios [7,9,10,11], and an absolute correlation exists between the molar or atomic O/C ratio and the biochar stability, with a ratio less than 0.2 indicating an estimated half-life of more than 1000 years [5]. Biochar stability also has a positive correlation with the fixed carbon content but a negative correlation with the ash content [11,12]. Additionally, the effects of alkali and alkaline earth metallic (AAEM) species on biochar stability exist [13,14,15,16,17], which can alter the reaction pathways between the chars and O_2_ during the pyrolysis, exerting effects on the char structure [15]. Therefore, in this study, proximate and ultimate analyses were performed on the biochar C structure, including fixed carbon (FC) content, ash content determination, and carbon–hydrogen–oxygen–nitrogen elemental composition measurements [5,8,9,11,18]. As of late, researchers have utilized the Van Krevelen diagram with the H/C atomic ratio against the O/C ratio to explore the development of aromaticity and aromatization in biochar [19]. Fourier transform infrared (FTIR) and X-ray photoelectron spectroscopy (XPS) are the frequent techniques for detecting changes in the carbon- and oxygen-containing functional groups that can reflect the carbonization process and the oxidation resistance of pyrolyzed biochar [20,21,22,23,24,25]. Raman analysis can provide detailed information about aromatic carbon structures [20,23,26,27,28,29,30]. For instance, the stability of biochar is highly correlated with the presence of amorphous aromatic carbon structures with smaller ring systems or the cross-link degree produced by nonaromatic structures (such as alkyl-CHn-) [29,30,31,32,33,34]. Temperature is decisive among pyrolysis parameters for biochar structure evolution. The atomic O/C and H/C ratios in biochar decrease with temperature, whereas the FC and ash contents increase [35,36,37,38,39,40,41,42,43,44,45,46].

Investigations have also shown that temperature has a positive correlation with the aromaticity and aromatic condensation of biochar [7,19]. However, the effects of the feedstock types and holding time cannot be ignored. The most common feedstocks for biochar production are woody biomass from different tree species and agricultural biomass from grass or crop plants [36]. Under pyrolysis at temperatures between 400 and 525 °C, the atomic H/C ratio of the spruce-based biochar decreased from 1.04 to 0.47, whereas that of the poplar-derived biochar decreased from 0.79 to 0.41 [38]. The ash content of the former increased from 1.9% to 4.7%, while that of the latter increased from 3.5% to 6.8% [38]. Recently, some studies found that the effects of feedstock types on seemed to be more significant at lower temperatures (≤500 °C) but affected less or little at higher temperatures (600–900 °C) [39,40]. This result means that the variation in biochar structure with temperature can be limited by the feedstock types when pyrolysis happens at moderate temperatures (~900 °C). In addition, research on feedstocks has been mostly restricted to interspecific rather than intraspecific comparisons. Biochar from different parts of the same species differed in their properties about the aromatic carbon structure. For example, Douglas fir wood biochar showed an increase from 49.58% to 83.15% in its FC content and experienced an increase from 0.6% to 1.13% in its ash content after pyrolysis at temperatures between 400 and 650 °C. In contrast, the FC content in Douglas fir bark chars increased from 51.50% to 73.89% and the ash content increased from 4.72% to 8.85% [37]. Moreover, pine needle biochar made at 300–700 °C had a carbon content of 84.19–93.67% [41], which was greater than that (72–81%) in biochar from pine residues under pyrolysis temperatures between 350 and 700 °C [19]. Therefore, we selected different parts (the branch and leaf parts, separately) of the same tree species as raw materials to produce biochar for comparison and analysis.

Currently, a growing body of the literature has begun to explore the potential effects of holding or residence time. Some claimed that an extension of the holding period significantly enhanced the carbonization of biochar, decreased the content of organic matter, and reduced the risks from microbial attacks [42]. On the other hand, the holding time exerted no significant effects on the biochar structure at 550 °C [43]. However, this result contradicts the report that biochar produced from manure, crop residue, and municipal solid wastes at temperatures between 300 and 700 °C with longer heating time enjoyed an advantage in thermal–oxidative stability [42]. These results suggest that an interaction between temperature, holding time, and feedstock types needs to be explored. In other words, a mechanism could potentially exist in which temperature can be considered the focal predictive variable that affects biochar structure, which is the outcome variable, and the holding time, feedstock types, or both can affect the relationship between them. Meanwhile, feedstock types can also influence the effect of the holding time on the relationship between temperature and biochar structure. 

Such a mechanism can be explored by the moderation analysis. Moderation and mediation effect analysis was developed by Baron et al. in the 1980s [47,48] and has been widely used in management, psychology, clinics, and other disciplines [49,50,51,52]. In 2012, Hayes et al. proposed the conditional process analysis (CPA), which is an upgraded version of simple moderation and mediation analysis [53,54]. CPA aims to explain more complex mediation or moderation mechanisms, such as a moderated moderation in which the moderation of a variable on the relationship between the focal predictive variable and outcome variable could be influenced by another variable [54,55]. Therefore, CPA is suitable for studying the relationship between biochar structure and multiple pyrolysis parameters, providing a more systematic, complete, and detailed reference for biochar manufacturing in the future. The specific modeling and parameter settings are provided in Section 3 Materials and Methods.

Based on current knowledge, two aspects can be examined: the first is the moderating effects of the holding time and feedstock types (the branch or leaf feedstock) on the evolution of the biochar structure with respect to temperature, and the second is the difference between the moderating effects of holding time on the biochar structure of BC and LC. Thus, the focuses of our study are to (a) identify how the carbon structures of BC and LC evolve with temperature and time; (b) cluster the observed carbon structure characteristics into different aspects of biochar structure using principal component analysis (PCA); (c) assess biochar stability according to these aspects of biochar structure; and (d) explore the moderating effect of holding time and feedstock type (branch or leaf feedstock) on the variation in these aspects of biochar structure with temperature. 

## 2. Results and Discussion

### 2.1. Fixed Carbon and Ash Content Analysis

A positive correlation existed between the fixed carbon (FC) and ash contents of biochar samples and the temperature, even though the holding time changed (Figure 1a,b). Specifically, the FC content of BC increased from 17–36% to 70–80% when the temperature increased from 350 to 750 °C with a holding time from 0.5 to 2 h. This range was narrower for the LC, where the FC content increased from 40–41% to 56–62%. The increasing trend in the ash content with temperature and time was similar between BC and LC. However, the ash content of LC averaged approximately 13%, almost three times the average ash content of BC, which was approximately 4%. The multi-way ANOVA showed that feedstock types (*p <* 0.001), pyrolysis temperature (*p* < 0.001), and holding time (*p* < 0.001) significantly affect the variation in the FC and ash contents. There is a significant interaction (*p* < 0.001) between the three factors (Appendix A). 

### 2.2. Elemental Composition Analysis 

The carbon content of LC increased from approximately 60% to 65–70% with variations in the temperature and holding period, while the carbon content of BC increased from 64–67% to 70–82% (Figure 1c). This result contrasted with the rapid decrease in the hydrogen and oxygen contents. The H contents of both BC and LC decreased approximately threefold, from approximately 4% to approximately 1% (Figure 1d). The O content of LC decreased from 25–26% to 16–17%, while that of BC decreased from 24–29% to 10–14% (Figure 1e). Notably, the N content in BC and LC minimally varied with temperature and time, but the nitrogen content in LC averaged approximately 1.9%, more than three times that in BC (Figure 1f). This difference can be explained by the higher N content (leaf:1.42%; branch:0.31%) of the leaf parts, as detailed information can be seen in Section 3 Material and Methods. 

A van Krevelen (VK) diagram was generated (Figure 2) to determine the change in aromaticity of the biochar with respect to temperature and time. The significance of temperature was considerable. The branch characterized a similar atomic H/C ratio to that of the leaf (between 1.6 and 1.7), indicating a high concentration of aliphatic compounds in the biomass. Pyrolysis at 350 °C caused a reduction in the H/C ratios by almost half, with approximate values of 0.8 in BC and 0.9 in LC. As pyrolysis continued to increase from 450 to 750 °C, the H/C ratio for BC decreased from 0.56 to approximately 0.16, while that of LC decreased from 0.66 to 0.21, indicating a sharp increase in the aromaticity of chars in this temperature range [8,10,56]. In addition, the atomic ratios of O/C in the branch and leaf parts remained similar, at approximately 0.6–0.7; then, these ratios decreased to approximately 0.3 in both chars with pyrolysis at 350 °C. At 450 °C, the O/C ratio of BC began to be evidently lower than that of LC. As the temperature increased to a range of 650–750 °C, the O/C ratio of BC decreased to approximately 0.1, almost half that of LC (approximately 0.2); thus, a separation of the two groups were observed in the VK plot in this temperature range. These results indicated a quicker decrease in the O/C ratio of BC, and the lower O/C and H/C ratios caused BC to outperform LC in terms of the aromaticity and stability of its carbon structure [5,7,8,9]. 

Regardless of the change in holding time, the H/C ratio was more than twice the O/C ratio for all biochar samples manufactured at 650 and 750 °C. Within this temperature range, the O/C ratios of BC and LC minimally changed, whereas their H/C ratios still decreased by approximately 27% and 17%, respectively. This potentially occurred because the considerable decreases in the O/C and H/C ratios at lower temperatures (~450 °C) were directly linked to the loss of H_2_O molecules in biomass and the decomposition of cellulose (full of OH and C-O groups), hemicellulose (with enormous C-O groups), and aliphatic compounds (rich in CH groups) [57,58]. However, the rapid loss of H atoms at high temperatures could be attributed to the further demethylation (such as the loss of methoxyl-O-CH_3_) in lignin and the enhanced polycondensations in the aromatic structures such as olefins [9,58,59,60,61]. 

### 2.3. FTIR Analysis of the Biochar Structure

FTIR was used to investigate how functional groups representing biochar structure evolved with temperature and time. Given the few changes in char structure with holding time, only the spectra for BC and LC recorded at four temperatures are displayed in Figure 3a,b, with the band assignments listed in Table 1.

Significant changes were observed in the 2800–3500 cm^−1^, 1000–1800 cm^−1^, and 700–900 cm^−1^ regions. Specifically, the band at 3427 cm^−1^, corresponding to the OH stretching vibration, decreased with temperature. The bands at 1700 cm^−1^ and 1315 cm^−1^, corresponding to carboxyl or carbonyl C=O stretching and C-O stretching vibrations, respectively, disappeared at temperatures above 450 °C. This was the result of direct dehydration, decarboxylation, and decarbonylating reactions during moderate-temperature (350 and 450 °C) pyrolysis. Additionally, the bands at 2921 and 2850 cm^−1^ were attributed to asymmetric and symmetric aliphatic CH_2_ (methylene) stretching, respectively, and were reduced after pyrolysis at 650 and 750 °C; this also corresponded to the fading of the band at 1435 cm^−1^ caused by the asymmetric deformation vibration of CH_2_ and CH_3_. However, the vibration at 1380 cm^−1^, attributed to the CH_3_ groups (methyl) in the aliphatic chains, was still present at temperatures up to 650 and 750 °C, although this peak was significantly reduced with the increase in temperature. These results supported the assumption that the two stages led to decreases in the atomic H/C ratio in the VK analysis. The first stage was mainly caused by the loss of the CH_2_ attached to the side chain of the aliphatic group, and the second was the release of CH_3_ caused by the breakage of the methoxy groups in lignin at high temperatures [57,61,63]. The band at 1614 cm^−1^, attributed to the aromatic ring stretching C=C or C=O vibrations, exhibited a considerable decrease between 450 and 650 °C. This result indicated that an “overlap” existed between the C=C and C=O bands due to the higher concentration of oxygen-containing functional groups in the low-temperature pyrolyzed biochar [25], and the substantial reduction above 450 °C indicated an increase in the loss of the O-aryl groups caused by the polycondensation at higher temperatures (≥400 °C) [64]. In addition, a peak at 1080 cm^−1^ emerged at temperatures between 650 and 750 °C, indicating that the increase in the degree of polymerization during pyrolysis stimulated the production of heterocyclic compounds [22,64]. 

In the fingerprint region between 700 and 900 cm^−1^, the intensity of the aromatic nucleus CH (2–3 adjacent H) at 770 cm^−1^ decreased considerably with temperature, whereas that of the aromatic nucleus CH (one adjacent H) at 870 cm^−1^ increased. The former reduction resulted from the cleavage of the groups attached to the edges of aromatic structures, while the latter increase was caused by the free radical transfer reaction between molecules or within molecules [23], indicating enhanced polycondensation in aromatic structures [21]. 

### 2.4. Raman Analysis of the Biochar Structure 

In Figure 3c, the D band (disorder or defect [59,65]) of BC and LC gradually shifted to the lower wavenumber side (from 1380 to 1350 cm^−1^), while the G band (graphite [59,65]) shifted slightly to a higher wavenumber (from 1590 to 1595 cm^−1^) as the temperature increased from 350 to 750 °C; these results indicated increasing crystallinity of the chars with temperature [27,28,65]. The Raman spectra were further deconvoluted and fitted by the second derivative using Origin Pro 2022 (Origin Lab, Northampton, MA, USA) with its nine Gaussian bands [27] (Figure 3d). The assignments for each band are listed in Table 2. A semiquantitative assessment of the changes in biochar structure with respect to temperature and time was performed according to the intensity ratios (band area percentage) of several of the main Raman bands (Figure 3e,f).

Pyrolysis from 350 to 750 °C led to a decrease in the content of small polyaromatic structures (D_R_ and G_L_) and oxygenated groups (G_R_), in contrast with the large increase in the proportion of the aromatic structures (D) with six or more rings (Appendix A). Therefore, the intensity ratio of I_D_/I_G_, which is a Raman parameter characterizing graphite-like carbon structures in coals [65], rapidly increased. This ratio for BC increased from 0.28–0.44 to 1.11–1.12, while the LC ratio increased from 0.46–0.68 to 1.62–1.84 (Figure 3e,f). The I_D_/I_G_ ratio is determined by the area ratio of the D band to the G band and decreases as the crystallinity increases [27,28,65,66]. The increase in this ratio in BC and LC meant that their carbon structures were highly disordered due to the presence of various kinds of aromatic rings; these structures developed because the dehydrogenation of the hydrocarbons and the expansion of the aromatic rings during pyrolysis were not yet crystallized into graphite [59,67,68,69,70]. The D_R_ and G_L_ bands that occupied the “overlap” region between the D and G bands indicated an aromatic ring system with 3–5 fused benzene rings; thus, the I_D_/I_(DR+GL)_ intensity ratio represents a medium-sized aromatic ring system with 3–6 benzene rings [62,65,71]. The I_D_/I_(DR+GL)_ ratio is determined by the ratio of the area of the D band to the sum of the band areas of the D_R_ and G_L_ bands. Here, the I_D_/I_(DR+GL)_ ratio rapidly increased with temperature (Figure 3e,f), indicating a higher amount of the 3–6 benzene ring systems in BC and LC because of incomplete polymerization of small aromatic compounds grafted with alkyl groups at temperatures lower than 900 °C [65]. This result indicated that medium-sized aromatic ring systems continued to condense into an increasingly large and stable aromatic cyclic system. Notably, the content of aryl–alkyl ether groups (D_L_) in BC and LC increased with temperature (Appendix A), contradicting the usual decreasing trend in oxygen-containing groups during carbonization. This was explained by the increase in heterocyclic compounds that were also found during the pyrolysis of brown coals [22] and rice husks [64] and also corresponded to the increase in the band at 1080 cm^−1^ observed via FTIR analysis (Figure 3a,b).

The cross-linking density and substitutional groups in carbonaceous materials can be briefly measured by the intensity ratio of I_S_/I_G_ (the area ratio of the S band to the G band), because the S band represents the sp2-sp3 carbon structures, such as C-C structures from aromatics, alkyls, or C-H structures attached to aromatic rings [26,27,29,65,67]. The I_S_/I_G_ ratios of BC and LC increased as the temperature increased from 350 to 650 °C despite change in the holding period, whereas the ratio minimally changed or slightly decreased between 650 and 750 °C (Figure 3e,f). The initial increase was attributed to the rapid formation of the alkyl–aryl C-C bonds caused by cross-linking reactions following decarboxylation reactions or the removal of other O-containing functional groups [59,65]. By the end of decarboxylation and deoxygenation reactions, which mainly occurred at temperatures lower than 500 °C [58], the production of the crosslinked structures leveled off or stopped, which could account for the slight or minimal changes in the I_S_/I_G_ ratios in BC and LC between 650 and 750 °C. 

When comparing BC and LC, the average I_D_/I_G_, I_D_/I_(DR+GL)_, and I_S_/I_G_ ratios of the former were 0.75, 0.95, and 0.21, while those for the latter were approximately 1.14, 1.26, and 0.29, respectively. Therefore, BC outperformed LC in terms of the degree of crystallinity, but LC featured greater cross-linking in carbon structures than BC.

### 2.5. XPS Analysis of the Biochar Structure 

Curve fitting of the high-resolution C1S XPS spectra was performed to provide additional detailed information about the carbon-and-oxygen-containing groups of chars, and five peaks were identified in the spectra of all the chars (Figure 4a). 

In particular, peak 1 was attributed to the aromatic or graphitic C-C/C=C (284.3~284.7 eV) [65,72,73] and accounted for the largest proportion of the total peak area. Peaks 2, 3, 4, and 5 corresponded to the hydrocarbon or aromatic C-H groups (285.2~285.6 eV) [72,73], the C-O/C-OH groups from alcohol or phenol (286.2~286.8 eV) [62,63,71,74], the carbonyl C=O (287.2~287.5 eV) [62,63], and the C=O/O-C=O groups in carboxylic or ester groups (288.8~289.4 eV) [63,74,75]. Moreover, the curve-fitting of the O1s XPS spectrum showed five peaks (Figure 4b). Peak I was the inorganic oxygen (530.3~530.8 eV) [71]; peak II was associated with the carbonyl C=O in quinines or ketones (531.3~531.8 eV) [62,63,72,74]; peak III represented the C-O or O-C=O in esters and anhydrides, or the oxygen atoms in -OH groups (532.1~532.6 eV) [62,63,72]; peak IV indicated the C-O in esters and anhydrides (533.3~533.8 eV) [64,73]; and peak V was attributed to the O-C=O in carboxyl groups (534.3~534.5 eV) [62,74].

The relative contents of each chemical component in the C1_S_ and O1_S_ spectra were calculated by deconvolution using Gaussian–Lorentzian fitting, and the variations in the relative content with pyrolysis temperature and time are shown in Figure 4c,d, respectively. Pyrolysis at 350~750 °C led to the decomposition of aliphatic compounds and demethylation of lignin, with a decrease in peak 2 in the C1_S_ spectra, indicating a reduction in the content of hydrocarbons. As the temperature increased, the sustained deoxygenation led to a large-scale breakage of the C-O bonds, as observed by a decrease in the intensity of peak 3 in the C1s spectra and peaks III and IV in the O1s spectra; as a result, a large number of -C- free radicals were released, and these species could form C=C or other functional groups with the stable structures by thermal polycondensation and cyclization. These results also corresponded to decreases in the FTIR bands attributed to the CH_2_ (at 2921 and 2850 cm^−1^) and C-O groups (at 1315 cm^−1^). Furthermore, the sustained dehydrogenation and deoxygenation following decarboxylation caused the O-C=O groups in the chars to initially increase and then decrease, as observed in the fluctuations in peak 5 in the C1_S_ spectra and peak IV in the O1_S_ spectra. Correspondingly, peak 1 in the C1_S_ spectra increased considerably with temperature, indicating the increasing proportion of aromatic carbon in the chars. However, a paradox was observed: the gradual increase in the contents of carbonyl C=O groups with temperature was inconsistent with the decrease in the intensity of the bands attributed to C=O groups at 1700 cm^−1^ in the FTIR spectra. This occurred because the FTIR band at 1700 cm^−1^ originated from the conjugated aromatic carbonyl or carboxyl C=O groups, including the oxygen atoms in esters, anhydrides, and carboxyl groups, which are beyond the reference functional groups of the C1_S_ peak 4 and O1_S_ peak II in the XPS data.

A comparison of the wide spectra of BC (Figure 4e) and LC (Figure 4f) clearly revealed more characteristic peaks, representing the presence of alkali and alkaline earth metallic species (AAEMs), including K, Ca, Mg, and Si, which were clearly identified on the surface of the LC samples. This could explain the higher ash contents in LC because K, Na, Ca, Mg, and Si ions are common ash-forming elements in lignocellulose biochar [13]. Alkaline earth metals, such as Ca and Mg, usually exist as organic compounds in the form of carboxylates that can bond with the carbon matrix via oxygen atoms under pyrolysis and increase the degree of cross-linking in the char [15,62,65]; this corresponded to the higher I_S_/I_G_ ratio in LC than in BC according to Raman analysis (Figure 3e,f).

### 2.6. Principal Component Analysis

Principal component analysis (PCA) was performed to provide a more comprehensive understanding and evaluation of the biochar structure and stability. The data used came from the following: the contents of ash, FC, and N; the atomic ratios of O/C and H/C; and the Raman parameters, such as the intensity ratios of I_D_/I_G_, I_D_/I_(DR+GL)_, and I_S_/I_G_. Bartlett’s test of sphericity and the Kaiser–Meyer–Olkin measure of sample adequacy indicated that the data were acceptable and suitable for factor analysis, with significance levels of 0.666 and 0.000, respectively. The results revealed that all carbon structure indicators were primarily classified into two components, which explained 81.122% of the total variance. The content of FC, the atomic ratios of O/C and H/C, and the intensity ratios of I_D_/I_G_ and I_D_/I_(DR+GL)_ were classified into principal component 1 (PC1) and accounted for 51.362% of the overall variation. Principal component 2 (PC2) mainly consisted of the contents of the ash and N and the intensity ratio for I_S_/I_G_ and accounted for 29.760% of the total variances. 

Table 3 gives the component scores of the 24 biochar types. Considering the positive and negative effects of the different indicators on the stability of biochar, all data were standardized using interquartile range normalization before scoring. The properties of PC1 directly influence the aromatic nature of the biochar structure and could be used to determine the stability of the chars. For PC1, BC and LC exhibited maximum values at 750 °C for 2 h, with the former occurring at 1.057 and the latter occurring at 0.900. The PC2 attributes correspond to the nonaromatic nature of biochar; these could affect biochar stability in other ways. For example, increases in the ash content led to the reductions in the FC content [12], higher amounts of N and S weakened the biochar stability [7,8,11,74,75], and an increased IS/IG ratio indicated an increase in the cross-linking structures that could prevent the development of larger aromatic ring systems [27,65,76]. Therefore, the scores of all chars in PC2 generally decreased with pyrolysis temperature and time due to the increasing trend in the N and ash content and in the ratios of IS/IG. When the two-side effects of PC1 and PC2 on biochar stability were considered, both BC and LC attained the highest total score (TS) under pyrolysis at 750 °C for 2 h, with a maximum TS of 97.603 for BC and 61.700 for LC. Additionally, the stability score of BC averaged 62.14, while the stability score of the LC averaged 33.70. This also indicated that the carbon structure was more stable in BC than in LC.

### 2.7. Conditional Process Analysis 

The outcome variable *Y* in Model 3 came from the results of the PCA analysis. PC1 represented the aromatic nature of biochar (ARO), which included the FC content, the atomic O/C and H/C ratios, and the I_D_/I_G_ and I_D_/I_(DR+GL)_ intensity ratios of chars. PC2 represented the nonaromatic nature (NARO), which involved the N and ash contents and the intensity ratio of I_S_/I_G_. Table 4 and Appendix A show the results. Here, the moderator *W*, which is the difference between branch and leaf feedstock, is labeled “Part” for simplicity.

#### 2.7.1. Simple Effects of Pyrolysis Temperature, Holding Time, and Feedstock Types on Biochar Structures

In Table 4, the ARO and NARO of biochar considerably increased with temperature (*β_Tempt_* for ARO = 0.005, *p* < 0.001; *β_Tempt_* for NARO = 0.001, *p* < 0.05), but modestly decreased with time (*β_Time_* for ARO = -0.198, *p* > 0.05; *β_Time_* for NARO = −0.260, *p* > 0.05). This result is consistent with the considerable variation in biochar structures with respect to temperature and the mild changes with holding hours. Meanwhile, the ARO and NARO of LC were obviously greater than BC (*β_Part_* for ARO = −0.881, *p* < 0.01; *β_Part_* for NARO = −1.791, *p* < 0.001). This could be explained by the higher H/C, O/C, I_D_/I_G_, I_D_/I_(DR+GL)_, and I_S_/I_G_ ratios in LC than in BC (Figure 2 and Figure 3e,f). The N and ash contents in LC were also greater than those in BC (Figure 1b,f). However, the aromaticity and stability of biochar decreased with the increases in the ratios of O/C and H/C and in the contents of N and ash. Moreover, the increasing I_D_/I_G_ and I_D_/I_(DR+GL)_ ratios represented a growth in amorphous aromatic carbons. These results led to an increased aromaticity but a decreased degree of order in aromatic structures within the molecular structure of chars [27]. Therefore, these results did not contradict the lower scores of LC in stability assessment in the previous PCA analysis.

#### 2.7.2. Moderating Roles of Holding Time and Feedstock Types on Biochar Structures

Table 4 also indicates that both the holding time and feedstock types (Part) had a significant moderating effect on the variation in aromatic structures of biochar with temperature (*β*_X×M_ for ARO = 0.001, *p* < 0.05; *β*_X×W_ for ARO = 0.002, *p* < 0.001). Moreover, the moderations of holding time on the variation in aromatic structures concerning temperature were significantly different between BC and LC (*β*_X×M×W_ for ARO = −0.001, *p* < 0.05). Specifically, LC exhibited a more drastic increasing trend in aromatic structures with temperature as the holding time extended from 0.5 to 2 h. In contrast, an extension of pyrolysis time affected less or slightly reduced the increase in aromatic structures of BC with respect to temperature (Appendix A). The reason may be that longer holding hours increased the ash content in BC, which was unfavorable for an increase in aromatic structures (Figure 1b). Another reason may be a counterbalance between the effects of the pyrolysis temperature and the kiln heating time [7]. In VK analysis, the change in atomic ratios caused by holding time became weaker as the temperature increased from 650 to 750 °C (Figure 2). Furthermore, the aromatic structure in BC increased considerably faster with temperature than that in LC under the same pyrolysis time (Appendix A). This result corresponded to the faster decreases in the atomic O/C ratio of BC when the temperature increased from 450 to 750 °C (Figure 2). 

Comparatively, the holding time and feedstock types (Part) had no significant moderating effects on the variation in the nonaromatic structure of biochar with temperature (*β*_X×M_ for NARO = 0.000, *p* > 0.05; *β*_X×W_ for NARO = −0.001, *p* > 0.05). No significant differences could be found between BC and LC (β_X×M×W_ for NARO = 0.000, *p* > 0.05) when comparing the moderating effects of the holding time. Appendix A supports this result. An extension of the holding period minimally enhanced the increases in nonaromatic structures of BC and LC with respect to temperature. In addition, although the nonaromatic structure in LC increased a little more quickly than that in BC with temperature, this difference is not statistically significant. This result is consistent with the similar variation in nonaromatic structures in BC and LC with temperature and time in previous discussions. 

#### 2.7.3. Differences in Nonaromatic Structures between the Branch- or Leaf-Based Biochar

Based on these results, the most considerable differences caused by feedstock type (branch and leaf feedstock) can be pointed out. These consist in the nonaromatic structure of BC and LC. The N (1.52–2.16%) and ash (10.03–17.43%) contents of the LC samples almost tripled compared to those (N: 0.41–0.79%; ash: 1.06–6.75%) of the BC samples. The XPS analysis showed that LC contained more AAEM species (K, Ca, and Mg) than the BC samples (Figure 4e,f). This result led to a higher I_S_/I_G_ intensity ratio in LC, indicating a greater cross-linked degree in its carbon structure.

The N content in most wood-based biochar (WBC) under pyrolysis between 200 and 900 °C is usually between 0.30 and 1.00% [19,36,37,38,39,40]. Comparatively, a higher N content can be found in biochar obtained from pine needles (3.64–4.1%) [41]. However, the biochar from pine wood had a lower N content of 0.24–0.44% [19]. Moreover, the ash content in the biochar from pine needle was 7.20–18.74% [41], whereas the biochar from pine wood was 1.70–3.00% [19]. Furthermore, the biochar from Douglas fir wood and bark had ash contents of 0.70–1.10% and 4.72–8.85%, respectively [37]. Therefore, different parts of the same tree species may cause a considerable difference in the nitrogen and ash contents of their biochar products. The difference in N and ash content could negatively influence the increase in the aromaticity of biochar and potentially lead to a less stable carbon structure in char samples [7,8,9]. This also explained why BC had a faster increase in aromatic structures with temperature and had better biochar stability than LC.

## 3. Materials and Methods

### 3.1. Biomass Collection and Biochar Preparation

*Taxodium ascendens* (TA) was selected as the feedstock for biochar production. Based on its wide utilization for greening and restoration in southeast China, the TA species have great potential for extensive application in the manufacture of daily chemicals and in the production of modified wood products as needed. Considering the tremendous amount of forest waste produced annually by TA trees in Anhui and Jiangsu provinces, their recycling for the creation of biochar products would benefit nearby environments. Additionally, our findings can provide insight into the potential of the TA species as target feedstocks for biochar production.

Chi Shan Lake Park, a national wetland park situated at the intersection of Anhui and Jiangsu Provinces (118°41′28.362″ E, 32°22′59.175″ N) provided the biomass needed to make char. After drying at 105 °C for 24 h, the biomass (branch or leaf parts) was ground to a powder with a particle size of less than 0.22 mm using an electrical grinder. Table 5 showed the feedstock characteristics. Within a quartz tube (tube size: length = 600 mm; outside diameter = 50 mm; wall thickness = 3 mm) furnace reactor (OTF-1200X-S, Kejing MTI, Hefei, Anhui, China), the branch- or leaf-based powder contained in a lidded ceramic crucible (crucible size: length = 60 mm; width = 30 mm; height = 4 mm) was pyrolyzed at 350, 450, 650, and 750 °C, holding for the periods of 0.5, 1, and 2 h at each target temperature (5 °C/min). Nitrogen gas flowed into the reactor to ensure an oxygen-free environment (70 mL/min), and the pyrolysis was performed in batches. The chars manufactured at the same temperature and for the same hours were combined, mixed, and gently ground to pass through 200 mesh sieves before being sealed into an airtight container for later measurements. The leaf-based chars were labeled as LC, and the branch-based chars as BC. Table 5 gives characteristics about the raw materials, i.e., TA branch and leaf parts.

### 3.2. Fixed Carbon, Ash, and Elemental Composition Analysis

Use a modified thermal analysis method [19,77] to analyze fixed carbon (FC) and ash content of biochar samples. In brief, each char sample contained in a ceramic crucible was dried for 24 h at 105 °C, heated for 1 h at 450 °C, and then heated for another 6 h at 750 °C, recording the weight loss between the three. Ash content was the weight of the residue after the third heating, while FC content was the difference between the initial weight and the three weight losses. Using the elemental analyzer (EA 2400 II, PerkinElmer, Waltham, MA, USA) to determine the contents of C, N, and H, while the O content was determined using the elemental analyzer (Varia ELIII, Elementa, Frankfurt, Germany) by combusting the samples at 1150 °C, both of which were carried out simultaneously on the same batch of samples. 

### 3.3. Spectrum Analysis

FTIR analysis was performed using a Nicolet iS20 FTIR (Thermo Fisher Scientific, Waltham, MA, USA). All spectra were obtained at a resolution of 4 cm^−1^ in the range of 400–4000 cm^−1^, with 32 scans for each spectrum. Raman analysis was performed on a confocal-micro-Raman spectrometer, LabRAM HR Evolution (Horiba Scientific, Longjumeau, France). The crystallographic structure of the chars was characterized using an excitation laser at 633 nm with a 50× objective focusing to a ~2 μm diameter laser spot on the samples. All spectra were recorded in the range of 500~2500 cm^−1^, with three points on each sample for scanning to ensure accurate spectrum information. The deconvolution of the Raman data was performed to determine the deformation of biochar during pyrolysis using the second derivative (Origin Pro) and a technique used by Zhang et al. to determine the number and positions of peaks in the original Raman spectra [27]. Full-spectrum scanning was used for X-ray photoelectron spectroscopy (XPS, ESCALAB250i Thermo Fisher Scientific, Waltham, MA, USA) to probe variations in the carbon-and-oxygen-containing functional groups on the biochar surface, with monochromator Al as a target source. The chemical states of C and O and the contents of these states were identified by narrow-band scanning XPS at 30 eV with a 0.1 step size, while C1s (284.8 eV) was selected as the internal standard to eliminate the effect of nuclear power. The deconvolution of the C1s and O1s XPS spectra was performed using Avantage Version 5.9931 software (Thermo Fisher Scientific, Waltham, MA, USA).

### 3.4. Statistical Analysis

#### 3.4.1. Principal Component Analysis 

The biochar structure and stability needed multiple indicators for their descriptions. They included the following: the atomic ratios of O/C and H/C; the contents of ash, FC, and N; and the Raman parameters, such as the intensity ratios of I_D_/I_G_, I_D_/I_(DR+GL)_, and I_S_/I_G_. Therefore, we used PCA to cluster these indicators into two or three aspects for subsequent stability level assessments and moderation tests. Principal component analysis was conducted using IBM SPSS (IBM, Armonk, NY, USA), version 22 statistics; this software was available as learning trials through the official platform. 

#### 3.4.2. Conditional Process Analysis 

In moderation analysis, the effect of a qualitative or quantitative variable and its influence on the direction or strength of the relationship between an independent or predictor variable and a dependent or outcome variable is explored and tested. The qualitative or quantitative variable is designated as the moderator variable [47,48]. The simplest form of the moderation is depicted conceptually in Figure 5a, and is statistically described in Figure 5b and the following linear Equation (1):
(1)$$Y=i_{\gamma }+b_{1}X+b_{2}M+b_{3}XM$$

Above, *X*, *Y*, and *M* represent the predictor, outcome, and moderator variables, respectively. The interaction term *XM* is considered the interaction effect of *X* and *M* [54]. The regression coefficients, including $b_{1}$, $b_{2}$, and $b_{3}$, can be estimated by including *X*, *M*, and *XM* as predictors of *Y* in Equation (1) [47,48,54,55]. The *p*-value for the weight for *XM* or a confidence interval, provided by standard regression analysis software IBM SPSS version 22 (IBM, Armonk, NY, USA), is used for inference as to whether *X*’s effect on *Y* is linearly moderated by *M*, or whether *X* and *M* interact in their influence on *Y* [48,52,54,55]; this also indicates that $b_{1}+b_{3\;}M$ is the conditional effect of *X* on *Y* [54,55]. 

Considering the multiple moderating effects of holding time and feedstock type in the previous aspect mentioned above, simple moderation analysis is not suitable. However, more than 76 CPA models [78] have been used to explore and explain the more complex moderation mechanisms involved. Among those models, Model 3 is suitable and was used in our study. The model conceptual diagram and statistical path plot are presented in Figure 5c,d, respectively. Here, the temperature was considered to be the focal predictor *X*, the part representing branch or leaf feedstock (branch = 1; leaf = 0), the holding time was considered to be the moderating variables *M* and *W*, and the biochar structure characteristics were considered to be the outcome variable *Y*. Modeling and testing were performed in PROCESS 3.5, a special plug-in for SPSS macros. The mechanism can be statistically described in the following equation—Equation (2): (2)$$Y=i_{\gamma }+b_{1}X+b_{2}M+b_{3}W+b_{4}XM+b_{5}XW+b_{6}MW+b_{7}XMW$$
The moderating effects of feedstock type (branch or leaf feedstock) and holding time were tested by the significance of coefficients $b_{4}$, $b_{5}$, and $b_{7}$, and the conditional effect of *X* on *Y* was calculated as $b_{1}+b_{4}M+b_{5}W+b_{7}MW$. 

## 4. Conclusions

The results obtained in this study showed that pyrolysis between 450 and 750 °C was the target stage for the aromatization of BC and LC; here, demethylation, depolymerization, and condensation accounted for the increase in aromatic carbon structures. Apart from the C-O bonds, the C-H bonds could also release free radicals and form stable aromatic ring structures by polycondensation and cyclization; this was shown as notable variation in the CH vibration bands in the FTIR spectrum and corresponded to a quicker decrease in the atomic H/C ratio according to van Krevelen analysis. Additionally, with higher FC concentrations (BC: 17.16–80.13%; LC: 39.14–62.23%), less N (BC: 0.41–0.79%; LC: 1.52–2.16%) and ash (BC:1.06–6.95%; LC: 10.03–17.43%) contents, lower atomic ratios of O/C (BC:0.09–0.34; LC:0.18–0.32) and H/C (BC: 0.15–0.85; LC: 0.20–0.91), and lower intensity ratios of I_D_/I_G_ (BC:0.37–1.12; LC: 0.68–1.84), I_D_/I_(DR+GL)_ (BC:0.27–2.26; LC: 0.39–2.84), and I_S_/I_G_ (BC:0.04–0.37; LC:0.18–0.41), BC showed more improved aromaticity, crystalline degree, and cross-linked structure than LC under the same pyrolysis conditions. This result was consistent with the PCA stability assessment, which showed that BC had an average score of 62.14, and LC had an average score of 33.70. 

According to Model 3 from the CPA analysis, the temperature and feedstock type (the branch or leaf feedstock) had significant simple effects on the aromatic and nonaromatic structures in biochar. In addition, the holding time and feedstock type significantly moderated the variation in aromatic structure of biochar with temperature. In contrast, there was no considerable moderating effects from holding time or feedstock type on the change in nonaromatic structure (the N and ash contents and the I_S_/I_G_ intensity ratio) of biochar with respect to temperature, that is, because the N and ash contents and the I_S_/I_G_ ratio in BC and LC were more influenced by the difference between branch and leaf feedstocks. Their variations with pyrolysis temperature and time were similar. Therefore, branch and leaf separation during biochar manufacturing is necessary and advisable. Slow pyrolysis (1 h) at temperatures between 350 and 750 °C can improve the stability of biochar derived from the branch and leaf parts of *Taxodium ascendens.* Besides, the leaf-based biochar had a higher nitrogen concentration. This finding suggests that the leaf-derived biochar has greater potential to raise the soil’s total N content and improve plant uptake of N, both of which can increase the yields of crops such as wheat, rice, and grains [79,80,81].

## Figures and Tables

**Figure 1 plants-13-00460-f001:**
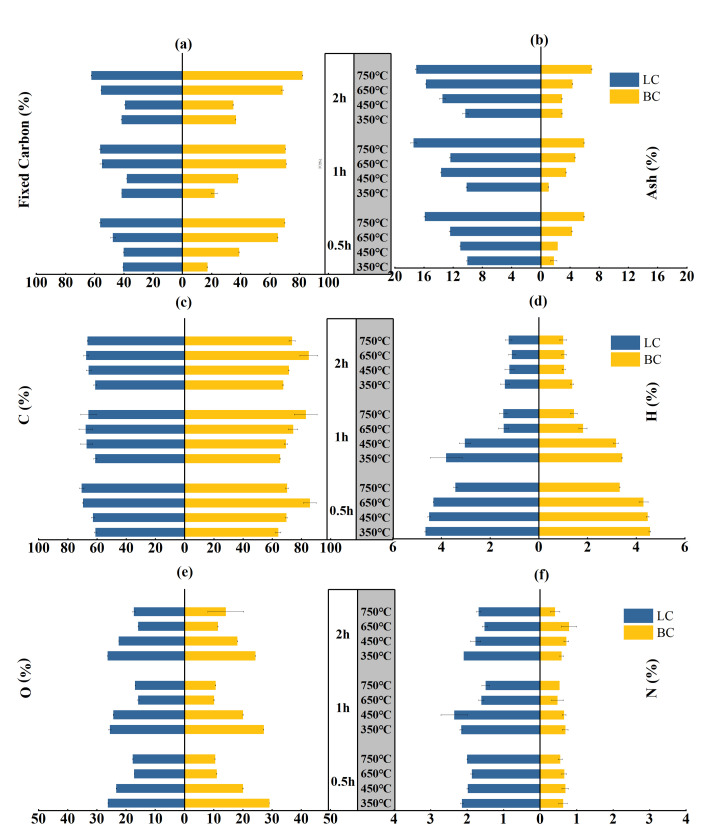
Fixed carbon, ash, C, H, O, and N contents of the BC and LC pyrolyzed by different temperatures (°C) and time durations (h): (**a**,**b**) are the contents of fixed carbon and ashs respectively; (**c**,**d**,**e**,**f**) are the contents of C, H, O and N, respectively.

**Figure 2 plants-13-00460-f002:**
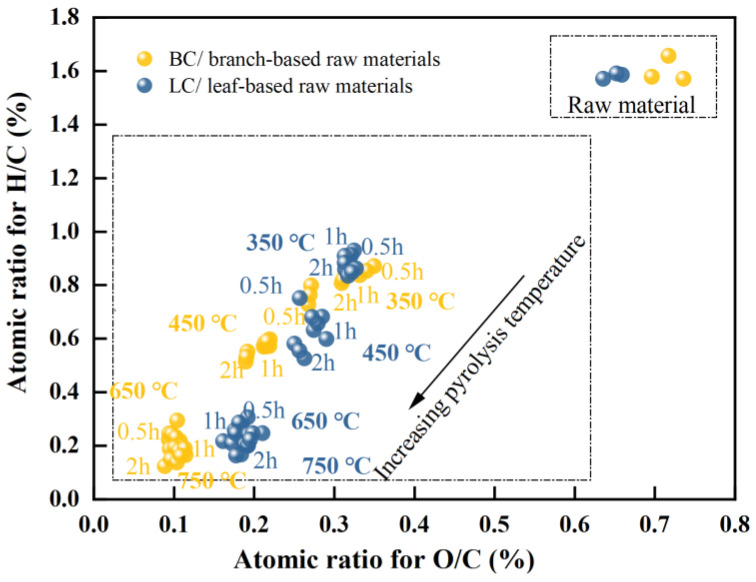
Van Krevelen plot for the branch-and-leaf-based raw materials (data points labeled as raw materials) and their biochar (data points with the labels of different temperature and time); the dotted frame is used to distinguish the raw material group from the biochar group.

**Figure 3 plants-13-00460-f003:**
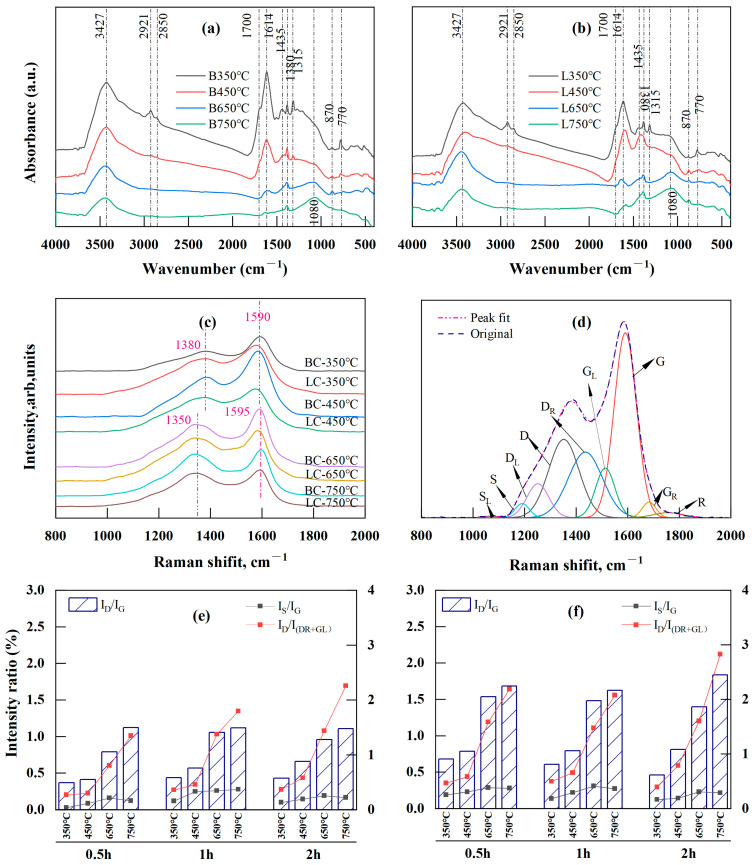
FTIR and Raman analysis results: (**a**,**b**) are FTIR spectra of BC (B) and LC (L), respectively; (**c**) is the positional shift of Raman spectra for BC and LC made at different pyrolysis temperatures; (**d**) is curve-fitting of the Raman spectra of chars; (**e**,**f**) are intensity ratios of I_D_/I_G_, I_S_/I_G_, and I_D_/I_(DR+GL)_ with temperature and time for BC and LC, respectively.

**Figure 4 plants-13-00460-f004:**
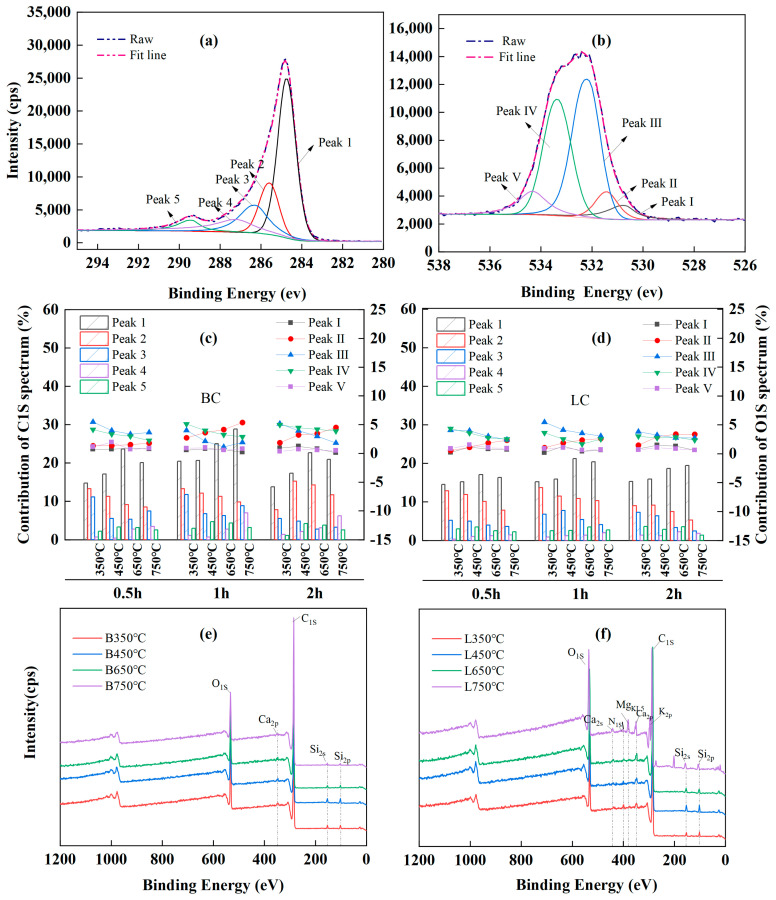
XPS analysis results: (**a**,**b**) peak deconvolution of XPS C1_S_ and O1_S_, respectively; (**c**,**d**) are relative content of functional groups in XPS C1_S_ and O1_S_ spectra, respectively; (**e**,**f**) are wide spectra of BC (B) and LC (L), respectively.

**Figure 5 plants-13-00460-f005:**
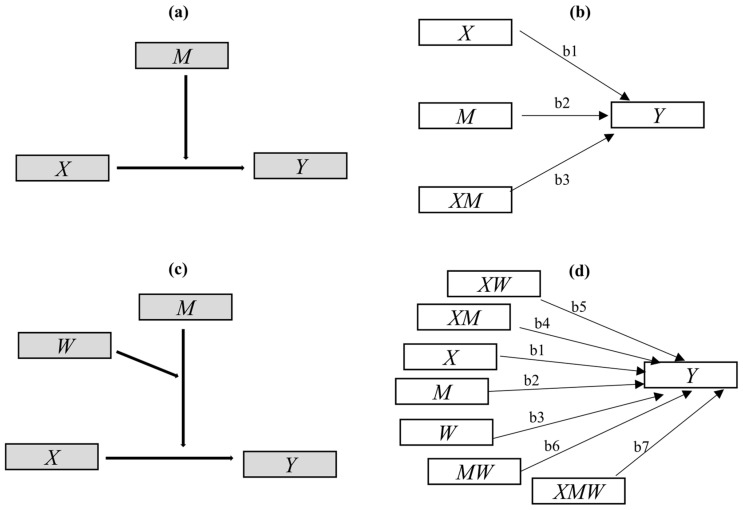
Diagrams for conditional process analysis (CPA) models: (**a**) conceptual diagram of the simplest moderation model; (**b**) its path plot in statistics, indicating the conditional effect of *X* on *Y* = b1 + b3*M*; (**c**) conceptual diagram of Model 3 in CPA; (**d**) the path plot of Model 3 in statistics, indicating the conditional effect of *X* on *Y* = b1 + b4*M* + b5*W* + b7*MW*.

**Table 1 plants-13-00460-t001:** Fourier-transform infrared spectroscopy band assignments.

Wavenumber (cm^−1^)	Assignments for Reference
3500–3200	O-H stretching vibration from acid [19,20,57]
2921	Asymmetric methylene CH_2_ stretching vibration in alkane [21,23]
2856	Symmetric methylene CH_2_ vibration in alkane [21,23]
1700	Carbonyl or carboxyl C=O stretching vibration [25]
1670–1600	Conjugated aromatic ring stretching C=C [57,62]
1435	CH_3_ and CH_2_ deformation vibration [19,24]
1380–1375	Aliphatic chains CH_3_- vibration [25]
1000–1350	C-O from organic alcohols, phenols, ethers, esters [19,25,57]
1000–1100	C-O stretching vibration in aliphatic ethers or alcohols [57,62]
870	Aromatic nucleus CH, one adjacent H deformation [21,23]
770	Aromatic nucleus CH, two to three adjacent H deformation [21,23]

**Table 2 plants-13-00460-t002:** Summary of Raman band assignments [27,59,62,65].

Band Name	Position (cm^−1^)	Description
D	1350	The ring breathing vibration in the graphite subunit; polycyclic aromatic hydrocarbon compounds (PAHs); aromatic with 6 or more rings
G	1590	Graphite; aromatic ring quadrant breathing; alkene C=C
D_R_	1435	Amorphous carbon structures, including methylene or methyl and analogous structures to PAHs; semi-circle breathing of aromatic rings
G_L_	1520	Aromatics with 3–5 rings; amorphous carbon structures
D_L_	1260	Aryl–alkyl ether; para-aromatics
G_R_	1696–1700	Carbonyl groups
S	1175–1185	The sp2-sp3 carbonaceous structures like C_aromatic_-C_alkyl_, aromatic (aliphatic) ethers, C-C on hydroaromatic rings, and C-H on aromatic rings
S_L_	1000–1100	C-H on aromatic rings and benzene (ortho-di-substitute) rings
R	1700–1800	Ester and lactone carbonyl vibrations

**Table 3 plants-13-00460-t003:** Component scores of the stability of 24 biochars, based on biochar C structure characteristics.

Score	Pyrolysis Temperature and Durations for Biochar Carbon Structure											
	350 °C			450 °C			650 °C			750 °C		
	0.5 h	1 h	2 h	0.5 h	1 h	2 h	0.5 h	1 h	2 h	0.5 h	1 h	2 h
BC												
S1	−0.693	−0.543	−0.337	−0.153	−0.083	−0.010	0.590	0.800	0.767	0.810	0.920	1.057
S2	1.907	1.733	1.797	1.853	1.630	1.673	1.627	1.480	1.483	1.483	1.480	1.457
TS	21.283	23.643	36.033	47.410	44.170	49.397	78.693	85.223	83.560	85.743	91.263	97.603
LC												
S1	−0.433	−0.460	−0.497	−0.197	−0.200	−0.027	0.573	0.633	0.660	0.687	0.790	0.900
S2	0.963	1.073	1.170	0.933	0.747	0.880	0.600	0.737	0.693	0.507	0.567	0.527
TS	6.367	8.373	9.343	17.820	11.857	24.843	47.247	54.427	54.677	50.297	57.317	61.700

S1 and S2, respectively, refer to the scores for principal component 1 and component 2; TS refers to total score.

**Table 4 plants-13-00460-t004:** Moderation tests of the holding time and feedstock types on the relationship between biochar structure and temperature using the conditional process analysis model 3.

Outcome Variable Y	Focal Predictor X and Moderators M and W	*β*	*t*-Value	Bootstrap = 5000	
				Confidence Interval 95%	
				Boot LLCI	Boot ULCI
ARO	Constant	−2.973	*** −14.216	−3.368	−2.696
	Tempt (X)	0.005	*** 13.180	0.004	0.006
	Time (M)	−0.198	−1.253	−0.461	0.107
	Part (W)	−0.881	** −2.978	−1.371	−0.257
	X × M	0.001	* 2.191	0.000	0.001
	X × W	0.002	*** 3.774	0.001	0.003
	X × M × W	−0.001	* −2.043	0.002	0.000
R^2^	0.976				
*F*	370.568				
NARO	Constant	0.422	1.645	−0.156	1.018
	Tempt (X)	0.001	* 2.659	0.000	0.002
	Time (M)	−0.260	−1.3392	−0.601	0.203
	Part (W)	−1.791	*** −4.932	−2.543	−0.973
	X × M	0.000	0.8213	−0.001	0.001
	X × W	−0.001	−1.1274	−0.002	0.001
	X × M × W	0.000	−0.2672	−0.001	0.001
R^2^	0.964				
*F*	242.756				

Tempt represents temperature; β, R^2^, and *F* represent the non-standard coefficient, R-square, and degree of freedom of the linear regressions in CPA Model 3; ARO represents the aromatic structure of biochar, while NARO represents the nonaromatic structure; Boot LLCI and Boot ULCI represents the lower and upper limits of confidence interval in Bootstrap tests; *p* < 0.05 *, *p* < 0.01 **, *p* < 0.001 ***.

**Table 5 plants-13-00460-t005:** Ultimate and proximate analyses of branch and leaf parts of *Taxodium ascendens*.

Feedstock	Proximate Analysis (mass %)				Ultimate Analysis (mass %)			
	Moisture	VM	FC	Ash	C	H	O	N
Branch	2.028	77.182	18.289	2.500	47.712	5.883	46.791	0.313
Leaf	1.931	76.113	17.343	4.623	47.234	6.182	40.000	1.424

VM means volatile matter; FC means fixed carbon.

## Data Availability

Data are contained within the article and Appendix A.

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
