# Peer review of "Using the Conditional Process Analysis Model to Characterize the Evolution of Carbon Structure in Taxodium ascendens Biochar with Varied Pyrolysis Temperature and Holding Time"

_plants, 2024, doi:10.3390/plants13030460_

Round 1
Reviewer 1 Report
Comments and Suggestions for Authors
I have keenly gone through the current manuscript titled “Using the Conditional Process Analysis Model to Characterize the Evolution in Carbon Structure of Biochar Derived from Different Parts of Taxodium ascendens under Varied Pyrolysis Temperature and Holding Time”. The authors have done a good extent of work. But, in my opinion some major revisions need to be made for the manuscript to comply with the standards set by the journal, before it could be considered for publication.
ABSTRACT:
L 13- 15, the author are suggested to use simpler sentences for better understanding of the readers.
L 21-23, the authors might consider a refinement of the linguistics to avoid confusion. Short, compact sentences help readers to stay focussed on results.
L 26, ‘around 0.2’ should be replaced with exact numbers.
L 26-27, Abbreviations create confusions; ‘loswer’ is a too qualitative a term to choose for results; a statistical insight would be appreciated.
L30, do the authors mean ‘a change in’ before ‘holding period’? If so, please add numerical information to catch reader attention.
INTRODUCTION:
The brevity of the introduction is appreciated. However, L61-72 should be considered for a change in the style of writing.
The novelty and objective behind this study is not very clear from this part. Although, in later sections the authors mention the research objectives, the writing style makes it difficult for the reader to figure out the total picture. It is suggested to rewrite the later paragraphs.
L 83-85, do not use whether for stating a hypothesis.
MATERIALS AND METHODS:
L 105, More details on instrumentation would be aprreciated.
L 142, The need behind performing PCA is missing
L 157- 160 remove ‘was’
L174 add figure number for reader convenience.
RESULTS AND DISCUSSIONS
Section 3.1, Are the differences statistically significant? The necessary numerics should be added. (L 188-191)
L 293, L 327; a little insight on these intensity ratios and their calculation would be helpful
CONCLUSION
Include more numbers. Conclusion lacks numerical information. It should be more compact.
Comments on the Quality of English Language
Moderate editing of English language required
Reviewer 2 Report
Comments and Suggestions for Authors
The study titled “Using the Conditional Process Analysis Model to Characterize the Evolution in Carbon Structure of Biochar Derived from Different Parts of Taxodium ascendens under Varied Pyrolysis Temperature and Holding Time” has been reviewed. In my opinion, the concept of the study is commendable as to unravel the carbon structure of biomass-based biochar. The presentation of results is also fair. However, author need to make some notable corrections before the paper can be acceptable for publication. The changes include:
Title
The current title is too long, ambiguous and tending towards meaninglessness. I suggest a modification to “Using Conditional Process Analysis Model to Characterize the Evolution in Carbon Structure of Taxodium ascendens Biochar under Varied Temperature and Holding Time”
Keywords:
Most of the keywords are taken directly from the title while others are meaningless statements. This is not right. Keywords should all not be from the title but words or phrases that truly reflect the content of the work for appropriate classification and indexation purposes
Arrangements
Author should pay careful attention to arrangement of sections. For example: Results and Discussion came before statistic which was also wrongfully numbered as 4. Is that the correct order by the journal? Should statistical analysis be a separated suction or a sub-section?
Introduction
1. The statement of problems, aim and objectives needs to be clearly stated. The concluding statement of this section already made future projections and that is commendable
2. The section is characterized by very poor Grammar and this should be corrected
Methods
Good
Results
Information provided here are okay. However, authors should modify all table to include footnotes that clearly explains the meaning of abbreviations
Conclusion
Good
References
References are okay.
General comment
1. The entire manuscript should be thoroughly checked for English Language.
2. Formatting and correct arrangement of sections should be clearly spelt out
3. There is a need for authors to actually justify the essence of the research in the introductory section

Needs moderate revision
Reviewer 3 Report
Comments and Suggestions for Authors
Review attached.

.
Author Response
Pleas see the attachment.

Round 2
Reviewer 1 Report
Comments and Suggestions for Authors
Authors have incorporated all comments raised by reviewers. This manuscript has been sufficiently improved and may be accepted for publication.